# Variational Inference for Mahalanobis Distance Metrics in Gaussian Process Regression

**Michalis K. Titsias**
Department of Informatics
Athens University of Economics and Business
`mtitsias@aueb.gr`

**Miguel Lázaro-Gredilla**
Dpt. Signal Processing & Communications
Universidad Carlos III de Madrid - Spain
`miguel@tsc.uc3m.es`

## Abstract

We introduce a novel variational method that allows to approximately integrate out kernel hyperparameters, such as length-scales, in Gaussian process regression. This approach consists of a novel variant of the variational framework that has been recently developed for the Gaussian process latent variable model which additionally makes use of a standardised representation of the Gaussian process. We consider this technique for learning Mahalanobis distance metrics in a Gaussian process regression setting and provide experimental evaluations and comparisons with existing methods by considering datasets with high-dimensional inputs.

## 1 Introduction

Gaussian processes (GPs) have found many applications in machine learning and statistics ranging from supervised learning tasks to unsupervised learning and reinforcement learning. However, while GP models are advertised as Bayesian models, it is rarely the case that a full Bayesian procedure is considered for training. In particular, the commonly used procedure is to find point estimates over the kernel hyperparameters by maximizing the marginal likelihood, which is the likelihood obtained once the latent variables associated with the GP function have been integrated out (Rasmussen and Williams, 2006). Such a procedure provides a practical algorithm that is expected to be robust to overfitting when the number of hyperparameters that need to be tuned are relatively few compared to the amount of data. In contrast, when the number of hyperparameters is large this approach will suffer from the shortcomings of a typical maximum likelihood method such as overfitting. To avoid the above problems, in GP models, the use of kernel functions with few kernel hyperparameters is common practice, although this can lead to limited flexibility when modelling the data. For instance, in regression or classification problems with high dimensional input data the typical kernel functions used are restricted to have the simplest possible form, such as a squared exponential with common length-scale across input dimensions, while more complex kernel functions such as ARD or Mahalanobis kernels (Vivarelli and Williams, 1998) are not considered due to the large number of hyperparameters needed to be estimated by maximum likelihood. On the other hand, while full Bayesian inference for GP models could be useful, it is pragmatically a very challenging task that currently has been attempted only by using expensive MCMC techniques such as the recent method of Murray and Adams (2010). Deterministic approximations and particularly the variational Bayes framework has not been applied so far for the treatment of kernel hyperparameters in GP models.

To this end, in this work we introduce a variational method for approximate Bayesian inference over hyperparameters in GP regression models with squared exponential kernel functions. This approach consists of a novel variant of the variational framework introduced in (Titsias and Lawrence, 2010) for the Gaussian process latent variable model. Furthermore, this method uses the concept of a standardised GP process and allows for learning Mahalanobis distance metrics (Weinberger and Saul, 2009; Xing et al., 2003) in Gaussian process regression settings using Bayesian inference. In

the experiments, we compare the proposed algorithm with several existing methods by considering several datasets with high-dimensional inputs.

The remainder of this paper is organised as follows: Section 2 provides the motivation and theoretical foundation of the variational method, Section 3 demonstrates the method in a number of challenging regression datasets by providing also a comprehensive comparison with existing methods. Finally, the paper concludes with a discussion in Section 4.

## 2 Theory

Section 2.1 discusses Bayesian GP regression and motivates the variational method. Section 2.2 explains the concept of the standardised representation of a GP model that is used by the variational method described in Section 2.3. Section 2.4 discusses setting the prior over the kernel hyperparameters together with a computationally analytical way to reduce the number of parameters to be optimised during variational inference. Finally, Section 2.5 discusses prediction in novel test inputs.

### 2.1 Bayesian GP regression and motivation for the variational method

Suppose we have data $\{y_i, \mathbf{x}_i\}_{i=1}^n$, where each $\mathbf{x}_i \in \mathbf{R}^D$ and each $y_i$ is a real-valued scalar output. We denote by $\mathbf{y}$ the vector of all output data and by $\mathbf{X}$ all input data. In GP regression, we assume that each observed output is generated according to $y_i = f(\mathbf{x}_i) + \epsilon_i$, $\epsilon_i \sim \mathcal{N}(0, \sigma^2)$, where the full length latent function $f(\mathbf{x})$ is assigned a zero-mean GP prior with a certain covariance or kernel function $k_f(\mathbf{x}, \mathbf{x}')$ that depends on hyperparameters $\boldsymbol{\theta}$. Throughout the paper we will consider the following squared exponential kernel function

$$k_f(\mathbf{x}, \mathbf{x}') = \sigma_f^2 e^{-\frac{1}{2}(\mathbf{x}-\mathbf{x}')^T \mathbf{W}^T \mathbf{W}(\mathbf{x}-\mathbf{x}')} = \sigma_f^2 e^{-\frac{1}{2}||\mathbf{W}\mathbf{x}-\mathbf{W}\mathbf{x}'||^2} = \sigma_f^2 e^{-\frac{1}{2}d_{\mathbf{W}}^2(\mathbf{x}, \mathbf{x}')}, \tag{1}$$

where $d_{\mathbf{W}}(\mathbf{x}, \mathbf{x}') = ||\mathbf{W}\mathbf{x} - \mathbf{W}\mathbf{x}'||$. In the above, $\sigma_f$ is a global scale parameter while the matrix $\mathbf{W} \in \mathbf{R}^{K \times D}$ quantifies a linear transformation that maps $\mathbf{x}$ into a linear subspace with dimension at most $K$. In the special case where $\mathbf{W}$ is a square and diagonal matrix, the above kernel function reduces to

$$k_f(\mathbf{x}, \mathbf{x}') = \sigma_f^2 e^{-\frac{1}{2}\sum_{d=1}^D w_d^2(x_d-x_d')^2}, \tag{2}$$

which consists of the well-known ARD squared exponential kernel commonly used in GP regression applications (Rasmussen and Williams, 2006). In other cases where $K < D$, $d_{\mathbf{W}}(\mathbf{x}, \mathbf{x}')$ defines a Mahalanobis distance metric (Weinberger and Saul, 2009; Xing et al., 2003) that allows for supervised dimensionality reduction to be applied in a GP regression setting (Vivarelli and Williams, 1998).

In a full Bayesian formulation, the hyperparameters $\boldsymbol{\theta} = (\sigma_f, \mathbf{W})$ are assigned a prior distribution $p(\boldsymbol{\theta})$ and the Bayesian model follows the hierarchical structure depicted in Figure 1(a). According to this structure the random function $f(\mathbf{x})$ and the hyperparameters $\boldsymbol{\theta}$ are a priori coupled since the former quantity is generated conditional on the latter. This can make approximate, and in particular variational, inference over the hyperparameters to be troublesome. To clarify this, observe that the joint density induced by the finite data is

$$p(\mathbf{y}, \mathbf{f}, \boldsymbol{\theta}) = \mathcal{N}(\mathbf{y}|\mathbf{f}, \sigma^2 I)\mathcal{N}(\mathbf{f}|\mathbf{0}, \mathbf{K}_{\mathbf{f},\mathbf{f}})p(\boldsymbol{\theta}), \tag{3}$$

where the vector $\mathbf{f}$ stores the latent function values at inputs $\mathbf{X}$ and $\mathbf{K}_{\mathbf{f},\mathbf{f}}$ is the $n \times n$ kernel matrix obtained by evaluating the kernel function on those inputs. Clearly, in the term $\mathcal{N}(\mathbf{f}|\mathbf{0}, \mathbf{K}_{\mathbf{f},\mathbf{f}})$ the hyperparameters $\boldsymbol{\theta}$ appear non-linearly inside the inverse and determinant of the kernel matrix $\mathbf{K}_{\mathbf{f},\mathbf{f}}$. While there exist a recently developed variational inference method applied to Gaussian process latent variable model (GP-LVM) (Titsias and Lawrence, 2010), that approximately integrates out inputs that appear inside a kernel matrix, this method is still not applicable to the case of kernel hyperparameters such as length-scales. This is because the augmentation with auxiliary variables used in (Titsias and Lawrence, 2010), that allows to bypass the intractable term $\mathcal{N}(\mathbf{f}|\mathbf{0}, \mathbf{K}_{\mathbf{f},\mathbf{f}})$, leads to an inversion of a matrix $\mathbf{K}_{\mathbf{u},\mathbf{u}}$ that still depends on the kernel hyperparameters. More precisely, the $\mathbf{K}_{\mathbf{u},\mathbf{u}}$ matrix is defined on auxiliary values $\mathbf{u}$ comprising points of the function $f(\mathbf{x})$ at some arbitrary and freely optimisable inputs (Snelson and Ghahramani, 2006a; Titsias, 2009). While this kernel matrix does not depend on the inputs $\mathbf{X}$ any more (which need to be integrated out in the GP-LVM case), it still depends on $\boldsymbol{\theta}$, making a possible variational treatment of those parameters

intractable. In Section 2.3, we present a novel modification of the approach in (Titsias and Lawrence, 2010) which allows to overcome the above intractability. Such an approach makes use of the so-called standardised representation of the GP model that is described next.

## 2.2 The standardised representation

Consider a function $s(\mathbf{z})$, where $\mathbf{z} \in R^K$, which is taken to be a random sample drawn from a GP indexed by elements in the low $K$-dimensional space and assumed to have a zero mean function and the following squared exponential kernel function:

$$k_s(\mathbf{z}, \mathbf{z}') = e^{-\frac{1}{2}||\mathbf{z}-\mathbf{z}'||^2}, \tag{4}$$

where the kernel length-scales and global scale are equal to unity. The above GP is referred to as *standardised* process, whereas a sample path $s(\mathbf{z})$ is referred to as a standardised function. The interesting property that a standardised process has is that it does not depend on kernel hyperparameters since it is defined in a space where all hyperparameters have been neutralised to take the value one. Having sampled a function $s(\mathbf{z})$ in the low dimensional input space $\mathbf{R}^K$, we can deterministically express a function $f(\mathbf{x})$ in the high dimensional input space $\mathbf{R}^D$ according to

$$f(\mathbf{x}) = \sigma_f s(\mathbf{W}\mathbf{x}), \tag{5}$$

where the scalar $\sigma_f$ and the matrix $\mathbf{W} \in \mathbf{R}^{K \times D}$ are exactly the hyperparameters defined in the previous section. The above simply says that the value of $f(\mathbf{x})$ at a certain input $\mathbf{x}$ is the value of the standardised function $s(\mathbf{z})$, for $\mathbf{z} = \mathbf{W}\mathbf{x} \in \mathbf{R}^K$, times a global scale $\sigma_f$ that changes the amplitude or power of the new function. Given $(\sigma_f, \mathbf{W})$, the above assumptions induce a GP prior on the function $f(\mathbf{x})$, which has zero mean and the following kernel function

$$k_f(\mathbf{x}, \mathbf{x}') = \mathbb{E}[\sigma_f s(\mathbf{W}\mathbf{x})\sigma_f s(\mathbf{W}\mathbf{x}')] = \sigma_f^2 e^{-\frac{1}{2}d_{\mathbf{W}}^2(\mathbf{x},\mathbf{x}')}, \tag{6}$$

which is precisely the kernel function given in eq. (1) and therefore, the above construction leads to the same GP prior distribution described in Section 2.1. Nevertheless, the representation using the standardised process also implies a reparametrisation of the GP regression model where a priori the hyperparameters $\boldsymbol{\theta}$ and the GP function are independent. More precisely, one can now represent the GP model according to the following structure:

$$\begin{aligned} s(\mathbf{z}) &\sim \mathcal{GP}(0, k_s(\mathbf{z}, \mathbf{z}')), \ \ \boldsymbol{\theta} \sim p(\boldsymbol{\theta}) \\ f(\mathbf{x}) &= \sigma_f s(\mathbf{W}\mathbf{x}) \\ y_i &\sim \mathcal{N}(y_i | f(\mathbf{x}_i), \sigma^2), \ \ i = 1, \ldots, n \end{aligned} \tag{7}$$

which is depicted graphically in Figure 1(b). The interesting property of this representation is that the GP function $s(\mathbf{z})$ and the hyperparameters $\boldsymbol{\theta}$ interact only inside the likelihood function while a priori are independent. Furthermore, according to this representation one could now consider a modification of the variational method in (Titsias and Lawrence, 2010) so that the auxiliary variables $\mathbf{u}$ are defined to be points of the function $s(\mathbf{z})$ so that the resulting kernel matrix $\mathbf{K}_{\mathbf{u},\mathbf{u}}$ which needs to be inverted does not depend on the hyperparameters but only on some freely optimisable inputs. Next we discuss the details of this variational method.

## 2.3 Variational inference using auxiliary variables

We define a set of $m$ auxiliary variables $\mathbf{u} \in \mathbf{R}^m$ such that each $u_i$ is a value of the standardised function so that $u_i = s(\mathbf{z}_i)$ and the input $\mathbf{z}_i \in \mathbf{R}^K$ lives in dimension $K$. The set of all inputs $\mathbf{Z} = (\mathbf{z}_1, \ldots, \mathbf{z}_m)$ are referred to as inducing inputs and consist of freely-optimisable parameters that can improve the accuracy of the approximation. The inducing variables $\mathbf{u}$ follow the Gaussian density

$$p(\mathbf{u}) = \mathcal{N}(\mathbf{u}|\mathbf{0}, \mathbf{K}_{\mathbf{u},\mathbf{u}}), \tag{8}$$

where $[\mathbf{K}_{\mathbf{u},\mathbf{u}}]_{ij} = k_s(\mathbf{z}_i, \mathbf{z}_j)$ and $k_s$ is the standardised kernel function given by eq. (4). Notice that the density $p(\mathbf{u})$ does not depend on the kernel hyperparameters and particularly on the matrix $\mathbf{W}$. This is a rather critical point, that essentially allows the variational method to be applicable, and comprise the novelty of our method compared to the initial framework in (Titsias and Lawrence, 2010). The vector $\mathbf{f}$ of noise-free latent function values, such that $[\mathbf{f}]_i = \sigma_f s(\mathbf{W}\mathbf{x}_i)$, covary with the vector $\mathbf{u}$ based on the cross-covariance function

$$k_{f,u}(\mathbf{x}, \mathbf{z}) = \mathbf{E}[\sigma_f s(\mathbf{W}\mathbf{x})s(\mathbf{z})] = \sigma_f \mathbf{E}[s(\mathbf{W}\mathbf{x})s(\mathbf{z})] = \sigma_f e^{-\frac{1}{2}||\mathbf{W}\mathbf{x}-\mathbf{z}||^2} = \sigma_f k_s(\mathbf{W}\mathbf{x}, \mathbf{z}). \tag{9}$$

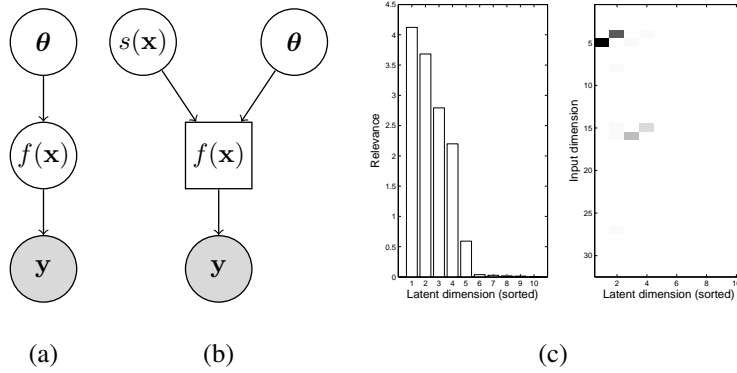

<div align="center">(a)        (b)            (c)</div>

Figure 1: The panel in (a) shows the usual hierarchical structure of a GP model where the middle node corresponds to the full length function $f(\mathbf{x})$ (although only a finite vector $\mathbf{f}$ is associated with the data). The panel in (b) shows an equivalent representation of the GP model expressed through the standardised random function $s(\mathbf{z})$, that does not depend on hyperparameters, and interacts with the hyperparameters at the data generation process. The rectangular node for $f(\mathbf{x})$ corresponds to a deterministic operation representing $f(\mathbf{x}) = \sigma_f s(\mathbf{Wx})$. The panel in (c) shows how the latent dimensionality of the `Puma` dataset is inferred to be 4, roughly corresponding to input dimensions 4, 5, 15 and 16 (see Section 3.3).

Based on this function, we can compute the cross-covariance matrix $\mathbf{K_{f,u}}$ and subsequently express the conditional Gaussian density (often referred to as conditional GP prior):

$$p(\mathbf{f}|\mathbf{u}, \mathbf{W}) = \mathcal{N}(\mathbf{f}|\mathbf{K_{f,u}}\mathbf{K_{u,u}^{-1}}\mathbf{u}, \mathbf{K_{f,f}} - \mathbf{K_{f,u}}\mathbf{K_{u,u}^{-1}}\mathbf{K_{f,u}^{T}}),$$

so that $p(\mathbf{f}|\mathbf{u}, \mathbf{W})p(\mathbf{u})$ allows to obtain the initial conditional GP prior $p(\mathbf{f}|\mathbf{W})$, used in eq. (3), after a marginalisation over the inducing variables, i.e. $p(\mathbf{f}|\mathbf{W}) = \int p(\mathbf{f}|\mathbf{u}, \mathbf{W})p(\mathbf{u})d\mathbf{u}$. We would like now to apply variational inference in the augmented joint model[1]

$$p(\mathbf{y}, \mathbf{f}, \mathbf{u}, \mathbf{W}) = \mathcal{N}(\mathbf{y}|\mathbf{f}, \sigma^2 I)p(\mathbf{f}|\mathbf{u}, \mathbf{W})p(\mathbf{u})p(\mathbf{W}),$$

in order to approximate the intractable posterior distribution $p(\mathbf{f}, \mathbf{W}, \mathbf{u}|\mathbf{y})$. We introduce the variational distribution

$$q(\mathbf{f}, \mathbf{W}, \mathbf{u}) = p(\mathbf{f}|\mathbf{u}, \mathbf{W})q(\mathbf{W})q(\mathbf{u}), \tag{10}$$

where $p(\mathbf{f}|\mathbf{u}, \mathbf{W})$ is the conditional GP prior that appears in the joint model, $q(\mathbf{u})$ is a free-form variational distribution that after optimisation is found to be Gaussian (see Section B.1 in the supplementary material), while $q(\mathbf{W})$ is restricted to be the following factorised Gaussian:

$$q(\mathbf{W}) = \prod_{k=1}^{K} \prod_{d=1}^{D} \mathcal{N}(w_{kd}|\mu_{dk}, \sigma_{kd}^2), \tag{11}$$

The variational lower bound that minimises the Kullback Leibler (KL) divergence between the variational and the exact posterior distribution can be written in the form

$$\mathcal{F} = \mathcal{F}_1 - \mathrm{KL}(q(\mathbf{W})||p(\mathbf{W})), \tag{12}$$

where the analytical form of $\mathcal{F}_1$ is given in Section B.1 of the supplementary material, whereas the KL divergence term $\mathrm{KL}(q(\mathbf{W})||p(\mathbf{W}))$ that depends on the prior distribution over $\mathbf{W}$ is described in the next section.

The variational lower bound is maximised using gradient-based methods over the variational parameters $\{\mu_{kd}, \sigma_{kd}^2\}_{k=1,d=1}^{K,D}$, the inducing inputs $\mathbf{Z}$ (which are also variational parameters) and the hyperparameters $(\sigma_f, \sigma^2)$.

## 2.4 Prior over $p(\mathbf{W})$ and analytical reduction of the number of optimisable parameters

To set the prior distribution for the parameters $\mathbf{W}$, we follow the automatic relevance determination (ARD) idea introduced in (MacKay, 1994; Neal, 1998) and subsequently considered in several models such as sparse linear models (Tipping, 2001) and variational Bayesian PCA (Bishop, 1999). Specifically, the prior distribution takes the form

$$p(\mathbf{W}) = \prod_{k=1}^{K} \prod_{d=1}^{D} \mathcal{N}(w_{kd}|0, \ell_k^2), \tag{13}$$

where the elements of each row of $\mathbf{W}$ follow a zero-mean Gaussian distribution with a common variance. Learning the set of variances $\{\ell_k^2\}_{k=1}^{K}$ can allow to automatically select the dimensionality associated with the Mahalanobis distance metric $d_{\mathbf{W}}(\mathbf{x}, \mathbf{x}')$. This could be carried out by either applying a Type II ML estimation procedure or a variational Bayesian approach, where the latter assigns a conjugate Gamma prior on the variances and optimises a variational distribution $q(\{\ell_k^2\}_{k=1}^{K})$ over them. The optimisable quantities in both these procedures can be removed analytically and optimally from the variational lower bound as described next.

Consider the case where we apply Type II ML for the variances $\{\ell_k^2\}_{k=1}^{K}$. These parameters appear only in the $\mathrm{KL}(q(\mathbf{W})||p(\mathbf{W}))$ term (denoted by KL in the following) of the lower bound in eq. (12) which can be written in the form:

$$\mathrm{KL} = \frac{1}{2} \sum_{k=1}^{K} \left[ \frac{\sum_{d=1}^{D} \sigma_{dk}^2 + \mu_{dk}^2}{\ell_k^2} - D - \sum_{d=1}^{D} \log \frac{\sigma_{dk}^2}{\ell_k^2} \right].$$

By first minimizing this term with respect to these former hyperparameters we find that

$$\ell_k^2 = \frac{\sum_{d=1}^{D} \sigma_{dk}^2 + \mu_{dk}^2}{D}, \quad k = 1, \dots, K, \tag{14}$$

and then by substituting back these optimal values into the KL divergence we obtain

$$\mathrm{KL} = \frac{1}{2} \sum_{k=1}^{K} \left[ \sum_{d=1}^{D} \log \sigma_{dk}^2 - D \log \left( \sum_{d=1}^{D} \sigma_{dk}^2 + \mu_{dk}^2 \right) + D \log D \right], \tag{15}$$

which now depends only on variational parameters. When we treat $\{\ell_k^2\}_{k=1}^{K}$ in a Bayesian manner, we assign inverse Gamma prior to each variance $\ell_k^2$, $p(\ell_k^2) = \frac{\beta^\alpha}{\Gamma(\alpha)} \left( \ell_k^2 \right)^{-\alpha-1} e^{-\frac{\beta}{\ell_k^2}}$. Then, by following a similar procedure as the one above we can remove optimally the variational factor $q(\{\ell_k^2\}_{k=1}^{K})$ (see Section B.2 in the supplementary material) to obtain

$$\mathrm{KL} = -\left( \frac{D}{2} + \alpha \right) \sum_{k=1}^{K} \log \left( 2\beta + \sum_{d=1}^{D} \mu_{kd}^2 + \sigma_{kd}^2 \right) + \frac{1}{2} \sum_{k=1}^{K} \sum_{d=1}^{D} \log(\sigma_{kd}^2) + \mathrm{const}, \tag{16}$$

which, as expected, has the nice property that when $\alpha = \beta = 0$, so that the prior over variances becomes improper, it reduces to the quantity in (15).

Finally, it is important to notice that different and particularly non-Gaussian priors for the parameters $\mathbf{W}$ can be also accommodated by our variational method. More precisely, any alternative prior for $\mathbf{W}$ changes only the form of the negative KL divergence term in the lower bound in eq. (12). This term remains analytically tractable even for priors such as the Laplace or certain types of spike and slab priors. In the experiments we have used the ARD prior described above while the investigation of alternative priors is intended to be studied as a future work.

## 2.5 Predictions

Assume we have a test input $\mathbf{x}_*$ and we would like to predict the corresponding output $y_*$. The exact predictive density $p(y_*|\mathbf{y})$ is intractable and therefore we approximate it with the density obtained by averaging over the variational posterior distribution:

$$q(y_*|\mathbf{y}) = \int \mathcal{N}(y_*|f_*, \sigma^2) p(f_*|\mathbf{f}, \mathbf{u}, \mathbf{W}) p(\mathbf{f}|\mathbf{u}, \mathbf{W}) q(\mathbf{u}) q(\mathbf{W}) df_* d\mathbf{f} d\mathbf{u} d\mathbf{W}, \tag{17}$$

where $p(\mathbf{f}|\mathbf{u}, \mathbf{W})q(\mathbf{u})q(\mathbf{W})$ is the variational distribution and $p(f_*|\mathbf{f}, \mathbf{u}, \mathbf{W})$ is the conditional GP prior over the test value $f_*$ given the training function values $\mathbf{f}$ and the inducing variables $\mathbf{u}$. By performing first the integration over $\mathbf{f}$, we obtain $\int p(f_*|\mathbf{f}, \mathbf{u}, \mathbf{W})p(\mathbf{f}|\mathbf{u}, \mathbf{W})d\mathbf{f} = p(f_*|\mathbf{u}, \mathbf{W})$ which yields as a consequence of the consistency property of the Gaussian process prior. Given that $p(f_*|\mathbf{u}, \mathbf{W})$ and $q(\mathbf{u})$ (see Section B.1 in the supplementary material) are Gaussian densities with respect to $f_*$ and $\mathbf{u}$, the above can be further simplified to

$$q(y_*|\mathbf{y}) = \int \mathcal{N}(y_*|\mu_*(\mathbf{W}), \sigma_*^2(\mathbf{W}) + \sigma^2)q(\mathbf{W})d\mathbf{W},$$

where the mean $\mu_*(\mathbf{W})$ and variance $\sigma_*^2(\mathbf{W})$ obtain closed-form expressions and consist of non-linear functions of $\mathbf{W}$ making the above integral intractable. However, by applying Monte Carlo integration based on drawing independent samples from the Gaussian distribution $q(\mathbf{W})$ we can efficiently approximate the above according to

$$q(y_*|\mathbf{y}) = \frac{1}{T}\sum_{t=1}^{T}\mathcal{N}(y_*|\mu_*(\mathbf{W}^{(t)}), \sigma_*^2(\mathbf{W}^{(t)}) + \sigma^2), \tag{18}$$

which is the quantity used in our experiments. Furthermore, although the predictive density is not Gaussian, its mean and variance can be computed analytically as explained in Section B.1 of the supplementary material.

## 3    Experiments

In this section we will use standard data sets to assess the performance of the proposed VDMGP in terms of normalised mean square error (NMSE) and negative log-probability density (NLPD). We will use as benchmarks a full GP with automatic relevance determination (ARD) and the state-of-the-art SPGP-DR model, which is described below. Also, see Section A of the supplementary material for an example of dimensionality reduction on a simple toy example.

### 3.1    Review of SPGP-DR

The sparse pseudo-input GP (SPGP) from Snelson and Ghahramani (2006a) is a well-known sparse GP model, that allows the computational cost of GP regression to scale linearly with the number of samples in a the dataset. This model is sometimes referred to as FITC (fully independent training conditional) and uses an active set of $m$ pseudo-inputs that control the speed vs. performance trade-off of the method. SPGP is often used when dealing with datasets containing more than a few thousand samples, since in those cases the cost of a full GP becomes impractical.

In Snelson and Ghahramani (2006b), a version of SPGP with dimensionality reduction (SPGP-DR) is presented. SPGP-DR applies the SPGP model to a linear projection of the inputs. The $K \times D$ projection matrix $\mathbf{W}$ is learned so as to maximise the evidence of the model. This can be seen simply as a specialisation of SPGP in which the covariance function is a squared exponential with a Mahalanobis distance defined by $\mathbf{W}^{\top}\mathbf{W}$. The idea had already been applied to the standard GP in (Vivarelli and Williams, 1998).

Despite the apparent similarities between SPGP-DR and VDMGP, there are important differences worth clarifying. First, SPGP's pseudo-inputs are model parameters and, as such, fitting a large number of them can result in overfitting, whereas the inducing inputs used in VDMGP are variational parameters whose optimisation can only result in a better fit of the posterior densities. Second, SPGP-DR does not place a prior on the linear projection matrix $\mathbf{W}$; it is instead fitted using Maximum Likelihood, just as the pseudo-inputs. In contrast, VDMGP does place a prior on $\mathbf{W}$ and variationally integrates it out.

These differences yield an important consequence: VDMGP can infer automatically the latent dimensionality $K$ of data, but SPGP-DR is unable to, since increasing $K$ is never going to decrease its likelihood. Thus, VDMGP follows Occam's razor on the number of latent dimensions $K$.

### 3.2    `Temp` **and** `SO`$_2$ **datasets**

We will assess VDMGP on real-world datasets. For this purpose we will use the two data sets from the WCCI-2006 Predictive Uncertainty in Environmental Modeling Competition run by Gavin

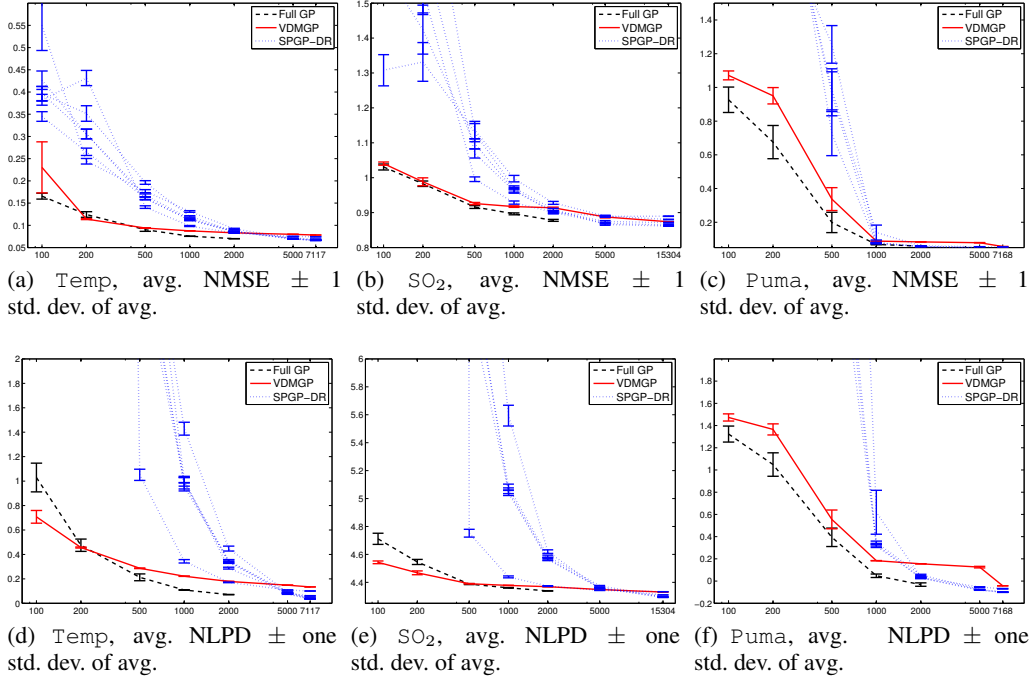

Figure 2: Average NMSE and NLPD for several real datasets, showing the effect of different training set sizes.

Cawley[2], called Temp and SO$_2$. In dataset Temp, maximum daily temperature measurements have to be predicted from 106 input variables representing large-scale circulation information. For the SO$_2$ dataset, the task is to predict the concentration of SO$_2$ in an urban environment twenty-four hours in advance, using information on current SO$_2$ levels and meteorological conditions.[3] These are the same datasets on which SPGP-DR was originally tested (Snelson and Ghahramani, 2006b), and it is worth mentioning that SPGP-DR's only entry in the competition (for the Temp dataset) was the winning one.

We ran SPGP-DR and VDMGP using the same exact initialisation for the projection matrix on both algorithms and tested the effect of using a reduced number of training data. For SPGP-DR we tested several possible latent dimensions $K = \{2, 5, 10, 15, 20, 30\}$, whereas for VDMGP we fixed $K = 20$ and let the model infer the number of dimensions. The number of inducing variables (pseudo-inputs for SPGP-DR) was set to 10 for Temp and 20 for SO$_2$. Varying sizes for the training set between 100 and the total amount of available samples were considered. Twenty independent realisations were performed.

Average NMSE as a function of training set size is shown in Figures 2(a) and 2(b). The multiple dotted blue lines correspond to SPGP-DR with different choices of latent dimensionality $K$. The dashed black line represents the full GP, which has been run for training sets up to size 2000. VD-MGP is shown as a solid red line. Similarly, average NLPD is shown as a function of training set size in Figures 2(d) and 2(e).

When feasible, the full GP performs best, but since it requires the inversion of the full kernel matrix, it cannot by applied to large-scale problems such as the ones considered in this subsection. Also, even in reasonably-sized problems, the full GP may run into trouble if several noise-only input dimensions are present. SPGP-DR works well for large training set sizes, since there is enough information for it to avoid overfitting and the advantage of using a prior on $\mathbf{W}$ is reduced. However,

Temp: 106 dimensions 7117/3558 training/testing data, SO$_2$: 27 dimensions 15304/7652 training/testing data.

[3]For SO$_2$, which contains only positive labels $y_n$, a logarithmic transformation of the type $\log(a + y_n)$ was applied, just as the authors of (Snelson and Ghahramani, 2006b) did. However, reported NMSE and NLPD figures still correspond to the original labels.

for smaller training sets, performance is quite bad and the choice of $K$ becomes very relevant (which must be selected through cross-validation). Finally, VDMGP results in scalable performance: It is able to perform dimensionality reduction and achieve high accuracy both on small and large datasets, while still being faster than a full GP.

### 3.3 `Puma` **dataset**

In this section we consider the 32-input, moderate noise version of the `Puma` dataset.[4] This is realistic simulation of the dynamics of a Puma 560 robot arm. Labels represent angular accelerations of one of the robot arm's links, which have to be predicted based on the angular positions, velocities and torques of the robot arm. 7168 samples are available for training and 1024 for testing.

It is well-known from previous works (Snelson and Ghahramani, 2006a) that only 4 out of the 32 input dimensions are relevant for the prediction task, and that identifying them is not always easy. In particular, SPGP (the standard version, with no dimensionality reduction), fails at this task unless initialised from a "good guess" about the relevant dimensions coming from a different model, as discussed in (Snelson and Ghahramani, 2006a). We thought it would be interesting to assess the performance of the discussed models on this dataset, again considering different training set sizes, which are generated by randomly sampling from the training set.

Results are shown in Figures 2(c) and 2(f). VDMGPR determines that there are 4 latent dimensions, as shown in Figure 1(c). The conclusions to be drawn here are similar to those of the previous subsection: SPGP-DR has trouble with "small" datasets (where the threshold for a dataset being considered small enough may vary among different datasets) and requires a parameter to be validated, whereas VDMGPR seems to perform uniformly well.

### 3.4 A note on computational complexity

The computational complexity of VDMGP is $\mathcal{O}(NM^2K + NDK)$, just as that of SPGP-DR, which is much smaller than the $\mathcal{O}(N^3 + N^2D)$ required by a full GP. However, since the computation of the variational bound of VDMGP involves more steps than the computation of the evidence of SPGP-DR, VDMGP is slower than SPGP-DR. In two typical cases using 500 and 5000 training points full GP runs in 0.24 seconds (for 500 training points) and in 34 seconds (for 5000 training points), VDMGP runs in 0.35 and 3.1 seconds while SPGP-DR runs in 0.01 and 0.10 seconds.

## 4 Discussion and further work

A typical approach to regression when the number of input dimensions is large is to first use a linear projection of input data to reduce dimensionality (e.g., PCA) and then apply some regression technique. Instead of approaching this method in two steps, a monolithic approach allows the dimensionality reduction to be tailored to the specific regression problem.

In this work we have shown that it is possible to variationally integrate out the linear projection of the inputs of a GP, which, as a particular case, corresponds to integrating out its length-scale hyperparameters. By placing a prior on the linear projection, we avoid overfitting problems that may arise in other models, such as SPGP-DR. Only two parameters (noise variance and scale) are free in this model, whereas the remaining parameters appearing in the bound are free variational parameters, and optimizing them can only result in improved posterior estimates. This allows us to automatically infer the number of latent dimensions that are needed for regression in a given problem, which is also not possible using SPGP-DR. Finally, the size of the data sets that the proposed model can handle is much wider than that of SPGP-DR, which performs badly on small-size data.

One interesting topic for future work is to investigate non-Gaussian sparse priors for the parameters $\mathbf{W}$. Furthermore, given that $\mathbf{W}$ represents length-scales it could be replaced by a random function $\mathbf{W}(\mathbf{x})$, such a GP random function, which would render the length-scales input-dependent, making such a formulation useful in situations with varying smoothness across input space. Such a smoothness-varying GP is also an interesting subject of further work.

**Acknowledgments**

MKT greatly acknowledges support from "Research Funding at AUEB for Excellence and Extroversion, Action 1: 2012-2014". MLG acknowledges support from Spanish CICYT TIN2011-24533.

## Footnotes

[1] The scale parameter $\sigma_f$ and the noise variance $\sigma^2$ are not assigned prior distributions, but instead they are treated by Type II ML. Notice that the treatment of $(\sigma_f, \sigma^2)$ with a Bayesian manner is easier and approximate inference could be done with the standard conjugate variational Bayesian framework (Bishop, 2006).

[2]Available at http://theoval.cmp.uea.ac.uk/~gcc/competition/

[4]Available from Delve, see `http://www.cs.toronto.edu/~delve/data/pumadyn/desc.html`.

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
