[Supplementary Material · supplementary.pdf]

# Supplementary material for:
# Variational Inference for Mahalanobis Distance Metrics in Gaussian Process Regression

## A    Dimensionality reduction on a toy dataset

In order to illustrate the capabilities of the discussed methods, we will consider here a toy dataset. More precisely, 25-dimensional input vectors $\{\mathbf{x}_n\}_{n=1}^{400}$ are generated, with each component being a random sample from a uniform distribution between 0 and 1. Then the associated outputs $\{y_n\}_{n=1}^{400}$ are generated by computing the sinc of a linear combination of the first 5 dimensions and finally adding zero-mean Gaussian noise of standard deviation 0.1. The resulting data is plotted on Figure 1, using the appropriate linear transformation for the horizontal axis (which obviously is not available to any of the tested methods).

Figure 1: Toy dataset, with input vectors projected onto a single dimension, according to the generative model.

Data is randomly split in two sets; 100 data points are used for training and 300 for testing. Ten independent splits of the data are used. Out of the 25 input dimensions, 20 were completely irrelevant for this task. Note that no dimension can be pruned using unsupervised linear dimensionality reduction methods (such as PCA), since all inputs have the same statistical properties and the concept of "relevant" only makes sense in connection with the labels.

Performance figures using a full GP, SPGP-DR and VDMGP are displayed in Table 1. The full GP performs worst. For most splits, no learning takes place and predictions are just the mean of the training data. We observed the NMSE dropping slightly below 1 for some splits in which it was able to correctly prune out several irrelevant dimensions, but this was more the exception than the norm, as reflected in the reported average NMSE. SPGP-DR was run with different choices for the number of latent dimensions, $K$. It becomes obvious that an appropriate choice of this parameter is critical for correct performance of SPGP-DR: selecting $K$ too small (5) or too big (20) almost completely prevents learning. This choice can only be made by using cross-validation, since the model likelihood is potentially ever increasing (in practice, the likelihood found for increasing values of $K$ may decrease due to the effect of local minima in the search, but these values are still useless for the selection of $K$).

Finally, VDMGP shows the best results overall both in terms on NMSE and NLPD, with a single run, and without resorting to any type of cross-validation. Simply maximizing the variational bound results in pruning. In our experiment, all latent dimensions were pruned except one, which agrees with the ground truth. The number of inducing variables (analogously, pseudo-inputs for SPGP-DR) was set to 20 in this experiment.

Table 1: Average and its standard deviation for test NMSE and NLPD on the toy dataset for the compared methods.

| MODEL | NMSE | NLPD |
|---|---|---|
| GP | $1.06\pm0.08$ | $+0.76\pm0.08$ |
| SPGP-DR $(K=5)$ | $1.28\pm0.23$ | $+95.4\pm24.4$ |
| SPGP-DR $(K=10)$ | $0.47\pm0.13$ | $+19.7\pm6.93$ |
| SPGP-DR $(K=15)$ | $0.88\pm0.09$ | $+1.99\pm0.70$ |
| SPGP-DR $(K=20)$ | $1.00\pm0.01$ | $+0.62\pm0.01$ |
| VDMGP $(K=20)$ | $0.09\pm0.00$ | $-0.54\pm0.01$ |

## B  Mathematical details

In this extra material, we briefly mention additional details that were not included in the main text due to lack of space.

### B.1  Details of the variational method

The $\mathcal{F}_1$ term of the variational lower bound takes the following analytical form

$$
\begin{aligned}
\mathcal{F}_1 &= -\frac{n}{2}\log(2\pi) - \frac{n-m}{2}\log\sigma^2 + \frac{1}{2}\log|\mathbf{K}_{\mathbf{u},\mathbf{u}}| \\
&\quad - \frac{1}{2}\log|\sigma^2\mathbf{K}_{\mathbf{u},\mathbf{u}} + \boldsymbol{\Psi}_2| - \frac{1}{2\sigma^2}\mathbf{y}^T\mathbf{y} \\
&\quad + \frac{1}{2\sigma^2}\mathbf{y}^T\boldsymbol{\Psi}_1(\sigma^2\mathbf{K}_{\mathbf{u},\mathbf{u}} + \boldsymbol{\Psi}_2)^{-1}\boldsymbol{\Psi}_1^T\mathbf{y} \\
&\quad - \frac{\psi_0}{2\sigma^2} + \frac{1}{2\sigma^2}\mathrm{tr}(\mathbf{K}_{\mathbf{u},\mathbf{u}}^{-1}\boldsymbol{\Psi}_2),
\end{aligned}
$$

where $\psi_0 = \langle\mathrm{tr}(\mathbf{K}_{\mathbf{f},\mathbf{f}})\rangle_{q(\mathbf{W})} = n\sigma_f^2$, $\boldsymbol{\Psi}_1 = \langle\mathbf{K}_{\mathbf{f},\mathbf{u}}\rangle_{q(\mathbf{W})}$ is a $m\times m$ matrix with elements

$$
\begin{aligned}
[\boldsymbol{\Psi}_1]_{ij} &= \int k_{f,u}(\mathbf{x}_i,\mathbf{z}_j)\prod_{k=1}^{K}\mathcal{N}(\mathbf{w}_k|\boldsymbol{\mu}_k,\Sigma_k) \\
&= \sigma_f\prod_{k=1}^{K}\int e^{-\frac{1}{2}(\mathbf{x}_i^T\mathbf{w}_k-z_{jk})^2}\mathcal{N}(\mathbf{w}_k|\boldsymbol{\mu}_k,\Sigma_k)d\mathbf{w}_k \\
&= \sigma_f\prod_{k=1}^{K}\frac{1}{(\mathbf{x}_i^T\Sigma_k\mathbf{x}_i+1)^{\frac{1}{2}}}e^{-\frac{(\boldsymbol{\mu}_k^T\mathbf{x}_n-z_{jk})^2}{2(\mathbf{x}_i^T\Sigma_k\mathbf{x}_i+1)}}.
\end{aligned}
$$

where $\Sigma_k$ is a diagonal matrix with elements in the diagonal the variational variances $(\sigma_{k1}^2,\ldots,\sigma_{kD}^2)$. The matrix $\boldsymbol{\Psi}_2 = \langle\mathbf{K}_{\mathbf{u},\mathbf{f}}\mathbf{K}_{\mathbf{f},\mathbf{u}}\rangle_{q(\mathbf{W})}$ is a $m\times m$ matrix with elements

$$
\begin{aligned}
[\boldsymbol{\Psi}_2]_{jj'} &= \sum_{i=1}^{n}\int k_{f,u}(\mathbf{x}_i,\mathbf{z}_j)k_{f,u}(\mathbf{x}_i,\mathbf{z}_{j'})q(\mathbf{W})d\mathbf{W} \\
&= \sigma_f^2 e^{-\frac{1}{4}\sum_{k=1}^{K}(z_{jk}-z_{j'k})^2} \\
&\quad \times \sum_{i=1}^{n}\prod_{k=1}^{K}\frac{1}{(2\mathbf{x}_i^T\Sigma_k\mathbf{x}_i+1)^{\frac{1}{2}}}e^{-\frac{(\boldsymbol{\mu}_k^T\mathbf{x}_i-\bar{z}_k)^2}{2\mathbf{x}_i^T\Sigma_k\mathbf{x}_i+1}},
\end{aligned}
$$

where $\bar{\mathbf{z}} = \frac{\mathbf{z}_j+\mathbf{z}_{j'}}{2}$.

The optimal variational distribution over the inducing variables $\mathbf{u}$ takes the Gaussian form

$$
\begin{aligned}
q(\mathbf{u}) &= \mathcal{N}(\mathbf{u}|\mathbf{K}_{\mathbf{u},\mathbf{u}}\left(\sigma^2\mathbf{K}_{\mathbf{u},\mathbf{u}} + \boldsymbol{\Psi}_2\right)^{-1}\boldsymbol{\Psi}_1^T\mathbf{y}, \\
&\quad \sigma^2\mathbf{K}_{\mathbf{u},\mathbf{u}}\left(\sigma^2\mathbf{K}_{\mathbf{u},\mathbf{u}} + \boldsymbol{\Psi}_2\right)^{-1}\mathbf{K}_{\mathbf{u},\mathbf{u}}).
\end{aligned}
$$

The terms $\mu_*(\mathbf{W})$ and $\sigma_*^2(\mathbf{W})$ that appear in the predictive density (see Section 2.5) are the following

$$\mu_*(\mathbf{W}) = \mathbf{k}_{f_*,\mathbf{u}} \left(\sigma^2 K_{\mathbf{u},\mathbf{u}} + \mathbf{\Psi}_2\right)^{-1} \mathbf{\Psi}_1^T \mathbf{y}$$

$$\sigma_*^2(\mathbf{W}) = \sigma_f^2 - \mathbf{k}_{f_*,\mathbf{u}} K_{\mathbf{u},\mathbf{u}}^{-1} \mathbf{k}_{f_*,\mathbf{u}}^T$$
$$+ \sigma^2 \mathbf{k}_{f_*,\mathbf{u}} \left(\sigma^2 K_{\mathbf{u},\mathbf{u}} + \mathbf{\Psi}_2\right)^{-1} \mathbf{k}_{f_*,\mathbf{u}}^T,$$

where $\mathbf{k}_{f_*,\mathbf{u}} = \mathbf{k}_s(\mathbf{W}\mathbf{x}_*, \mathbf{Z})$. As mentioned in Section 2.5, the mean and the variance of the predictive density can be computed analytically. Specifically, the mean is

$$\mu_* = \boldsymbol{\psi}_{*,1} \boldsymbol{\alpha},$$

where $\boldsymbol{\alpha} = \left(\sigma^2 K_{\mathbf{u},\mathbf{u}} + \mathbf{\Psi}_2\right)^{-1} \mathbf{\Psi}_1^T \mathbf{y}$ and $\boldsymbol{\psi}_{*,1} = \langle \mathbf{k}_{f_*,\mathbf{u}} \rangle_{q(\mathbf{W})}$ where the latter is computed analogously to the $\mathbf{\Psi}_1$ matrix of the lower bound. Similarly the predictive variance is

$$\sigma_*^2 = \operatorname{tr}\left(\left[\sigma^2 \left(\sigma^2 K_{\mathbf{u},\mathbf{u}} + \mathbf{\Psi}_2\right)^{-1} - K_{\mathbf{u},\mathbf{u}}^{-1} + \boldsymbol{\alpha}\boldsymbol{\alpha}^T\right] \mathbf{\Psi}_{*,2}\right)$$
$$+ \sigma_f^2 + \sigma^2 - \mu_*^2$$

where $\mathbf{\Psi}_{*,2} = \langle \mathbf{k}_{f_*,\mathbf{u}}^T \mathbf{k}_{f_*,\mathbf{u}} \rangle_{q(\mathbf{W})}$.

## B.2 Variational inference over the variances $\{\ell_k^2\}_{k=1}^K$

When we follow a full Bayesian approach for the variances of the Gaussian prior $p(\mathbf{W}|L)$, where $L = \{\ell_k^2\}_{k=1}^K$, we need to extend the variational distribution to include a factor $q(L)$. In such case, the negative KL divergence in the variational lower bound from eq. (12) of the paper takes the form

$$-\operatorname{KL} = \int q(\mathbf{W})q(L) \log \frac{p(\mathbf{W}|L)p(L)}{q(\mathbf{W})q(L)} d\mathbf{W} dL$$
$$= \int q(L) \log \frac{e^{\int q(\mathbf{W}) \log p(\mathbf{W}|L) d\mathbf{W}} p(L)}{q(L)} dL$$
$$- \int q(\mathbf{W}) \log q(\mathbf{W}) d\mathbf{W}.$$

The maximum with respect to the variational distribution $q(L)$ is achieved when we reverse Jensen's inequality so that

$$-\operatorname{KL} = \log \int e^{\int q(\mathbf{W}) \log p(\mathbf{W}|L) d\mathbf{W}} p(L) dL$$
$$- \int q(\mathbf{W}) \log q(\mathbf{W}) d\mathbf{W},$$

which can be computed analytically

$$-\operatorname{KL} = \left(\frac{D}{2} + \alpha\right) \sum_{k=1}^K \log\left(2\beta + \sum_{d=1}^D \mu_{kd}^2 + \sigma_{kd}^2\right)$$
$$- \frac{1}{2} \sum_{k=1}^K \sum_{d=1}^D \log(\sigma_{kd}^2) + \text{const},$$

where

$$\text{const} = -\sum_{k=1}^K \log \frac{\beta^\alpha \Gamma\left(\frac{D}{2} + \alpha\right)}{\Gamma(\alpha)} - \left(\frac{D}{2} + \alpha\right) K \log 2.$$