[Reviews · NeurIPS 2013]

Submitted by Assigned_Reviewer_6

The paper proposes a variational bound on the length scale parameters of square-exponential-kernel Gaussian process regression models. The main idea is to separate the function to be inferred into a "standardised" sample from a unit-length-scale square-exponential kernel, and a linear scaling map of that latent function, then to impose factorisation between these two objects via a variational bound.

The paper is well written. It uses clear language and provides a compact introduction to previous work. To my knowledge, the idea is novel. Length scales are a perennial problem in kernel regression models, so the promise of a lightweight, efficient and effective approximate probabilistic solution (as opposed to the existing alternative of MCMC sampling) is a significant contribution.

My main concern is the experimental evaluation. The experiments presented in Figure 2 do a good job of arguing that the new method improves on SPGP-DR. What they don't show is whether the new method is actually good at capturing posterior mass in the length scales. The paper's point would be strengthened by a comparison between the independent marginals (Eq. 11) inferred by the new method and those found by a "gold standard", e.g. from an MCMC sampler. The variational bound contains some strong factorization constraints (Eq. 11), and such are known to cause overconfidence. This may, or may not be a problem here.

I would be grateful if the authors could clarify the following points in the feedback:

* I understand, from line 293 that the "full GP" is type-II maximum likelihood, with the ARD kernel? I'm surprised that this model is consistently better than the probabilistic, Mahalonobis SPGP-DR on the puma dataset. Section 3.3 does not seem to explain this.

* lines 380 and lines 404-409 argue that the new method is fast. But none of the plots quote runtimes. Of course, runtime comparisons must be taken with a grain of salt, but they are still interesting. Could you quote a few numbers? How fast is VDMGP compared to the full GP and to SPGP-DR, for small and large datasets?

Finally, a minor point about the conclusion: You are suggesting, for future work, to use locally varying length scales in the kernel. This is not part of the paper so not a core point of my review, but as far as I understand, the generalization you propose does not actually give a valid covariance function. See Mark Gibbs' PhD thesis (Cambridge), page 18, and Section 3.10.3 (page 46 and following), which also gives ideas for alternate approaches.

Summary: A well-written paper about a relevant, interesting method. Experimental evaluation is limited.

Submitted by Assigned_Reviewer_7

In a GP regression model, the process outputs can be integrated over analytically, but this is not so for (a) inputs and (b) kernel hyperparameters. Titsias etal 2010 showed a very clever way to do (a) with a particular variational technique (the goal was to do density estimation). In this paper, (b) is tackled, which requires some nontrivial extensions of Titsias etal. In particular, they show how to decouple the GP prior from the kernel hyperparameters. This is a simple trick, but very effective for what they want to do. They also treat the large number of kernel hyperparameters with an additional level of ARD and show how the ARD hyperparameters can be solved for analytically, which is nice.

While not on the level of Titsias etal, I find this a very nice technical extension, tackling a hard problem. The paper is also very well written and clear. The way of getting Titsias etal to work is rather straightforward, but nothing wrong with that: exactly what is needed here.

Unfortunately, the experimental results are a bit unconvincing (while the experiments are well done). First, it is not stated how hyperpars are treated for "full GP". I assume for now that there is no W in full GP, that a simple Gaussian kernel is used -- please correct in response if wrong. There is no point in using sparse GP for n < 5000 or so, and in that regime full GP outperforms everything else. This is unfortunate, given that the motivation for the whole work is to show that integrating out hyperpars. helps. I also wonder why experiments for full GP only go to n=2000. Then, for larger n, the proposed method does not work well: the much simpler VDMGP works better or as well.

All in all, it would be important to motivate this work properly. If the goal is to have a scalable approximation that also does well on hyperpars., then maybe a larger dataset would have to be used. If the goal is to show improvements due to integrating out hyperpars on rather small sets, then the failure to improve upon "full GP" would have to be understood. After all, why not use more inducing inputs and see what you get? While technically beautiful, the trick of Titsias etal needs to be better understood in the context of vanilla GP regression, as it makes rather strong assumptions, and this work would be a good place to start.
Summary: Proposes a method for approx. integrating out kernel hyperparameters in a GP regression context, building on previous work by Titsias etal to integrate out inputs. Technically very interesting contribution, the experimental results (while well done) are somewhat disappointing.

Submitted by Assigned_Reviewer_8

The paper presents a method that allows to variationally integrate out kernel hyperparameters in Gaussian process regression. The idea is to a priori decouple these parameters from the GP mapping by employing an intermediate GP with fixed parameters which, when combined with a transformation of the model likelihood, results in an equivalent overall representation. Then the variational method of [1] is applied.

This was an interesting paper to read and contributes to the relevant literature. Although the proposed method heavily builds on [1] there are two main factors that make this paper non-trivial:
firstly, the presentation of the method is extensive and clear. Secondly and more importantly, the standardized GP trick that enables [1] to be applied is a very simple but also non-obvious idea.

The computations seem sound. Although the experiments are convincing, I would like to see some more comparisons that reveal the true benefit of the marginalization and the importance of selecting appropriate priors. In particular, could you compare your method on toy data with an MCMC approach? How important is to optimize in a fully Bayesian way (with the inv. Gamma priors) versus using ML for the prior's hyperparameters? It would be nice to see the difference in an experimental setting. By the way, which of the two optimizations (fully Bayesian vs ML) are you using for your experiments? I can't see it being mentioned anywhere.

One minor issue that might be interesting to comment is, why does VDMGP seem to perform slightly worse that SPGP-DR in large datasets? Is it because of the resulting complicated optimization space that does not allow the inducing points end up in a good global optimum? Also, how is the projection matrix initialized in 3.2 for both methods? Is it just a random initialization? How sensitive is the method to this initialization?

One other technical question is, why do you use MC to obtain the predictive distribution since you can analytically find the mean and variance? Does it make any difference in practice? If yes, how big, and why?

Minor typo in line 354 (repetition of "these").

[1] Titsias and Lawrence, Bayesian Gaussian process latent variable model, 2010
Summary: Overall a solid paper, technically sound and well written. It comes with convincing experiments although I would like to see some more comparisons. I expect this paper to be of interest in the NIPS community and thus vote for acceptance.
Author Feedback

Author rebuttal: We would like to thank the reviewers for their comments and provide them with some additional information and clarifications.


Assigned_Reviewer_6:

- Yes, you are correct about the "full GP" in the experimental section referring to ML-II with ARD kernel. This was explained at the beginning of Section 3. We will explicitly mention that ML-II inference is used.

- Regarding the Puma data set: The effect that you mention is actually expected. The Puma dataset has 32 input dimensions, but only 4 of them are actually useful for prediction. This is mentioned for instance in Snelson & Ghahramani (2006a). VMDGP detects this, see Fig. 1.(c), as well as the corresponding caption. The plot of matrix W shows that feature selection is happening and that the more complex linear projection is not actually needed. Therefore, the full GP with ARD kernel is a better prior for this data set, since it is flexible enough to allow irrelevant dimensions to be pruned out but not more flexible, and only 4 hyperparameters are actually tuned. On the other hand, SPGP-DR and VDMGP include a full linear projection as part of their model. When the number of training data is small, the additional flexibility of these models reduces its performance. Nonetheless, the advantage of VMDGP over SPGP-DR shows the benefit of approximately marginalising out the linear projection.

- To give a rough idea of the computational cost, we provide here the time required to compute the objective function during training (either the evidence or the variational bound) as well as its derivatives with respect to all free parameters for the SO2 experiment:

Full GP: 0.24 secs (for 500 training points) and 34 secs (for 5000 training points)
VDMGP: 0.35 secs (for 500 training points) and 3.1 secs (for 5000 training points)
SPGP-DR: 0.01 secs (for 500 training points) and 0.10 secs (for 5000 training points) [This has to be repeated for several values of K]

- Thanks for the pointer to Mark Gibbs' thesis. We are aware that simply replacing the constant lengthscale \ell of a squared exponential kernel with an arbitrary function \ell(x) might render it non positive semidefinite. However, this is not what we meant. Instead, we suggest to model the output as a standardised GP over a non-linear distortion of the inputs, parameterized by w(x). This results in k(x, x') = exp(- (w(x)*x - w(x')*x')^2 / 2), which is a valid covariance function for any arbitrary function w(x) (since it transfers the computations to the space of the standardized GP, always producing positive semidefinite matrices). We will clarify this in the final version and point to Mark Gibbs' thesis as source for alternative approaches.

Assigned_Reviewer_7:

- Yes, we confirm that no full projection matrix W is used for the full GP and that a standard ARD kernel is used. This is mentioned at the beginning of Section 3. Standard ML-II is used to select hyperparameters.

- The proposed model aims to improve over SPGP-DR and scale well for large amounts of training data. This is shown in the experiments, with VMDGP being much better than SPGP-DR when the amount of training data is small and keeping up with SPGP-DR when the amount of training data is big. It is unfortunate that on these datasets a ML-trained full (i.e. non-sparse) GP with ARD kernel (a simpler prior) performs better. This would not be the case for datasets in which a full linear projection was a better model. Also, note that for 100 data points the NLPD is better for VDMGP than for the full GP both on Temp and SO2, so the posterior probability of the proposed model seems to give more accurate predictions when we have limited amount of training data.

Assigned_Reviewer_8:

- In our experiments we are using vague inverse Gamma priors (with very small values of alpha and beta, of the order of 1e-3) on the variance of the elements of W. We forgot to mention that and will include it in the next version of this paper.

- We tried two initialisations for the posterior over the linear projection: Random and PCA. PCA seemed to work better in general, so we used it for the experiments.

- Even though we can analytically compute the posterior mean and variance, the posterior itself is not a Gaussian density, but instead it has heavier tails than a Gaussian. Therefore, in order to more accurately compute this predictive probability density of test data, it is better to use MC.

- Thank you, we corrected the typo.